# Neural Graphical Models

## Abstract

Graphs are ubiquitous and are often used to understand the dynamics of a system. Probabilistic Graphical Models comprising Bayesian and Markov networks, and Conditional Independence graphs are some of the popular graph representation techniques. They can model relationships between features (nodes) together with the underlying distribution. Although theoretically these models can represent very complex dependency functions, in practice often simplifying assumptions are made due to computational limitations associated with graph operations. This work introduces Neural Graphical Models (NGMs) which attempt to represent complex feature dependencies with reasonable computational costs. Specifically, given a graph of feature relationships and corresponding samples, we capture the dependency structure between the features along with their complex function representations by using neural networks as a multi-task learning framework. We provide efficient learning, inference and sampling algorithms for NGMs. Moreover, NGMs can fit generic graph structures including directed, undirected and mixed-edge graphs as well as support mixed input data types. We present empirical studies that show NGMs' capability to represent Gaussian graphical models, inference analysis of a lung cancer data and extract insights from a real world infant mortality data provided by CDC.

*Keywords*: Graphical models, Deep learning, Learning representations
*Software*:NGM code link (provided in Supplementary)

## 1 Introduction

Graphical models are a powerful tool to analyze data. They can represent the relationship between the features and provide underlying distributions that model functional dependencies between them. Probabilistic graphical models (PGMs) are quite popular and often used to describe various systems from different domains. Bayesian networks (directed acyclic graphs) and Markov networks (undirected graphs) are able to represent many complex systems due to their generic mathematical formulation Pearl (1988); Koller & Friedman (2009). These models rely on conditional independence assumptions to make representation of the domain and the probability distribution over it feasible.

Learning, inference and sampling are operations that make such graphical models useful for domain exploration. Learning, in a broad sense, consists of fitting the distribution function parameters from data. Inference is the procedure of answering queries in the form of marginal distributions or reporting conditional distributions with one or more observed variables. Sampling is the ability to draw samples from the underlying distribution defined by the graphical model. One of the common bottlenecks of graphical model representations is having high computational complexities for one or more of these procedures. Figuring out approximate algorithms or coming up with analytically favorable underlying distributions have been topics of interest to the research community for the past few decades.

In particular, various graphical models have placed restrictions on the set of distributions or types of variables in the domain. Some graphical models work with continuous variables only (or categorical variables only) or place restrictions on the graph structure (e.g., that continuous variables cannot be parents of categorical variables in a DAG). Other restrictions affect the set of probability distributions the models are capable of representing, e.g., to multivariate Gaussian.

Practically, for graphical models to be widely adoptable, the following properties are desired:

- Facilitate rich representations of complex underlying distributions.
- Support various relationship representations including directed, undirected, mixed-edge graphs.

- Fast and efficient algorithms for learning, inference and sampling.
- Direct access to the learned underlying distributions for analysis.
- Handle different input data types such as categorical, continuous, images, text, and generic embedding representations.

In this work we propose Neural Graphical Models (NGMs) that satisfy the aforementioned desiderata in a computationally efficient way. NGMs accept a feature dependency structure that can be given by an expert or learned from data. The dependency structure may have the form of a graph with clearly defined semantics (e.g., a Bayesian network graph or a Markov network graph) or an adjacency matrix. Note that the graph may be either directed or undirected. Based on this dependency structure, NGMs represent the probability function over the domain using a deep neural network. The parameterization of such a network can be learned from data efficiently, with a loss function that jointly optimizes adherence to the given dependency structure and fit to the data. Probability functions represented by NGMs are unrestricted by any of the common restrictions inherent in other PGMs. They also support efficient inference and sampling.

The rest of this paper is organized as follows: in Section 2 we briefly review work most closely related to ours, in Section 3 we introduce Neural Graphical Models including representation, learning, inference, sampling and handling of extended data types. We present experiments, both on synthetic and real-life data in Section 4 and Appendix B, discuss design considerations and limitations of our framework in Appendix A and close with conclusions and directions for future work in Section 5.

## 2 RELATED WORK

Probabilistic graphical models aim to learn the underlying joint distribution from which input data is sampled. Often, to make learning of the distribution computationally feasible, inducing an independence graph structure between the features helps. In cases where this independence graph structure is provided by a domain expert, the problem of fitting PGMs reduces to learning distributions over this graph. Alternatively, there are many methods traditionally used to jointly learn the structure as well as the parameters Heckerman et al. (1995); Spirtes & Meek (1995); Koller & Friedman (2009); Scanagatta et al. (2019) and have been widely used to analyse data in many domains Barton et al. (2012); Bielza & Larrañaga (2014); Borunda et al. (2016); Shrivastava et al. (2019a); Shrivastava (2020).

A few researchers explored discriminative PGMs, learning not joint probability distributions over a domain, but an approximation to a conditional distribution $P(Y|X - Y)$ where $Y$ is a pre-selected subset of $X$, typically in the context of undirected graphs. The best known are conditional random fields (CRF) Lafferty et al. (2001). Discriminative models are more flexible in ignoring complex dependencies between most of the variables in the domain and focusing on their impact on a small subset. They often have faster and more accurate inference, albeit restricted to the pre-selected set of variables. Generative models have higher bias – they make more assumptions about the form of the distribution. The bias helps with regularization and avoiding overfitting. However, generative models are poorer predictors than discriminative models. In this work, we attempt to combine the advantages of both methods by creating a discriminative model capable of predicting the value of any variable in a domain.

Recently, many interesting deep learning based approaches for DAG recovery have been proposed Zheng et al. (2018; 2020); Lachapelle et al. (2019); Yu et al. (2019). These works primarily focus on the structure learning but technically they are learning a probabilistic graphical model. These works depend on the existing algorithms developed for the Bayesian networks for the inference and sampling tasks. A parallel line of work combining graphical models with deep learning are Bayesian deep learning approaches: Variational AutoEncoders, Boltzmann Machines etc. (Wang & Yeung, 2020). The deep learning models have significantly more parameters than traditional Bayesian networks. Thus, using these deep graphical models for downstream tasks is computationally expensive and often impedes their adoption.

We would be remiss not to mention the technical similarities NGMs have with some recent research works. First, we found 'Learning sparse nonparametric DAGs' Zheng et al. (2020) to be the closest in terms of representation ability. In one of their versions, they model each independence structure with a different neural network (MLP). However, their choice of modeling feature independence

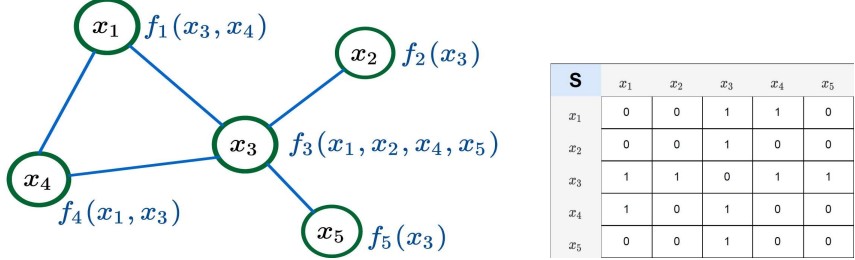

Figure 1: **Graphical view of NGMs**: The input graph **G** (undirected) for given input data $X \in \mathbb{R}^{M \times D}$. Each feature $x_i = f_i(\text{Nbrs}(x_i))$ is a function of the neighboring features. For a DAG, the functions between features will be defined by the Markov Blanket relationship $x_i = f_i(\text{MB}(x_i))$. The adjacency matrix (right) represents the associated dependency structure S.

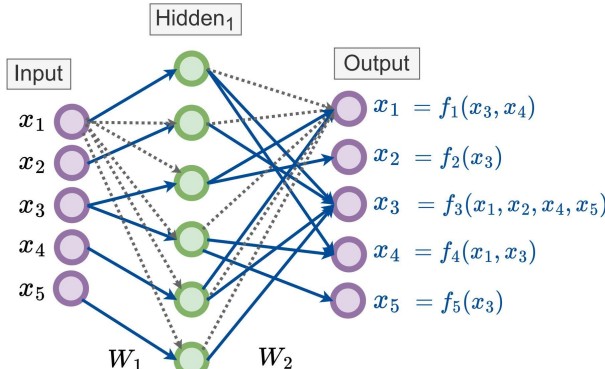

Figure 2: **Neural view of NGMs**: NN as a multitask learning architecture capturing non-linear dependencies for the features of the undirected graph in Fig. 1. If there is a path from the input feature to an output feature, that indicates a dependency between them. The dependency matrix between the input and output of the neural network reduces to a simple matrix multiplication operation $S_{nn} = \Pi_i |W_i| = |W_1| \times |W_2|$. Note that not all the zeroed out weights of the MLP (in black-dashed lines) are shown for the sake of clarity.

criterion differs from NGM. They zero out the weights of the row in the first layer of the NN to induce independence between the input and output features. This type of formulation restricts them from sharing the NNs across different factors. Second, we found similar path norm formulations of using the product of NN weights for input to output connectivity for NGMs in Lachapelle et al. (2019). They use the path norm to parametrize the DAG constraint for continuous optimization, while Shrivastava et al. (2020; 2022b) use the within unrolled algorithm framework to learn sparse gene regulatory networks.

There are methods that model the conditional independence graphs (Friedman et al., 2008; Belilovsky et al., 2017; Shrivastava et al., 2019b; 2022a) which are a type of graphical models that are based on underlying multivariate Gaussian distribution. Probabilistic Circuits (Peharz et al., 2020), Conditional Random Fields or Markov Networks (Sutton et al., 2012)are some other popular formulations. These PGMs often make simplifying assumptions on the underlying distributions and have certain restrictions on the input data type that can be handled. Real-world input data often consist of mixed datatypes (real, categorical, text, images etc.) and is challenging for the existing graphical model formulations to handle.

## 3   NEURAL GRAPHICAL MODELS

We propose a new probabilistic graphical model type, called Neural Graphical Models (NGMs) and describe the associated learning, inference and sampling algorithms. Our model accepts all input types and avoids placing any restrictions on the form of underlying distributions.

## 3.1 Problem setting

We are given input data $\mathbf{X}$ that have $M$ sample points with each sample consisting of $D$ features. An example of such data can be gene expression data, where data is a matrix of the microarray expression values (samples) and genes (features). Another example is a mix of continuous and categorical data describing a patient's health in a medical domain. We are also provided a graph $\mathbf{G}$ which can be directed, undirected or have mixed-edge types that represents our belief about the feature dependency relationships (in a probabilistic sense). Such graphs are often provided by experts and include inductive biases and domain knowledge about the underlying system functions. In cases where the graph is not provided, we make use of the state-of-the-art algorithms to recover DAGs or CI graphs, as described in Sec. 2. The NGM input is the tuple $(\mathbf{X}, \mathbf{G})$.

## 3.2 Representation

Fig. 1 shows a sample graph recovered and how we view the value of each feature as a function of the values of its neighbors. In the case of directed graphs, each feature's value is represented as a function of its Markov blanket in the graph. We use the graph $\mathbf{G}$ to understand the domain's dependency structure, but ignore any potential parametrization associated with it.

We introduce a 'neural' view which is another way of looking at $\mathbf{G}$, represented in Fig. 2. These neural networks are multi-layer perceptrons with appropriate input and output dimensions that represent graph connections in NGMs. Specifically, we view the neural networks as an 'open-box' and focus on the paths from input to output. These paths represent functional dependencies. Consider a neural network with H number of layers having ReLU non-linearity $f_{W_1,W_2,\cdots,W_H}(X_{\mathcal{I}}^k) = (\cdots (W_2 \cdot \mathrm{ReLU}(W_1 \cdot X_{\mathcal{I}}^k + b_1) + b_2) \cdots)$. The dimensions of the weights and biases are chosen such that the neural network input and output units are equal to $\mathcal{I}$. The product of the weights of the neural networks

---

**Algorithm 1:** NGMs: Learning algorithm

**Function** `proximal-init` $(X, S)$:
$\quad f_{\mathcal{W}} \leftarrow$ Init MLP using dimensions from S
$\quad f_{\mathcal{W}^0} \leftarrow$
$\quad\quad \arg\min_{\mathcal{W}} \sum_{k=1}^{M} \left\| X_{\mathcal{I}}^k - f_{\mathcal{W}}(X_{\mathcal{O}}^k) \right\|^2$
$\quad\quad$ (Using 'adam' optimizer for $E_1$ epochs)
$\quad$ **return** $f_{\mathcal{W}^0}$

**Function** `fit-NGM` $(X, S, f_{\mathcal{W}^0}, \lambda^0)$:
$\quad$ **For** $e = 1, \cdots, E_2$ **do**
$\quad\quad \mathcal{L}_{\mathrm{Lr}} = \sum_{k=1}^{M} \left\| X_{\mathcal{I}}^k - f_{\mathcal{W}^{e-1}}(X_{\mathcal{O}}^k) \right\|^2$
$\quad\quad\quad + \lambda^{e-1} \left\| \left( \Pi_i |W_i^{e-1}| \right) * S^c \right\|_1$
$\quad\quad \mathcal{W}^e \leftarrow$ backprop $\mathcal{L}_{\mathrm{Lr}}$ to update params
$\quad\quad \cdots$ (optional $\lambda$ update) $\cdots$
$\quad\quad \lambda^e \leftarrow \left\| \left( \Pi_i |W_i^e| \right) * S^c \right\|_2^2$
$\quad\quad$ Detach $\lambda^e$ from the computational graph
$\quad$ **return** $\Theta, Z, \lambda$

**Function** `NGM-learning` $(X, S)$:
$\quad f_{\mathcal{W}^0} \leftarrow$ `proximal-init` $(X, S)$
$\quad \lambda^0 \leftarrow \left\| \left( \Pi_i |W_i^0| \right) * S^c \right\|_2^2$
$\quad f_{\mathcal{W}} \leftarrow$ `fit-NGM` $(X, S, f_{\mathcal{W}^0}, \lambda^0)$
$\quad$ **return** $f_{\mathcal{W}}$

---

$S_{nn} = \Pi_i |W_i| = |W_1| \times |W_2| \times \cdots \times |W_C|$ gives us path dependencies. If $S_{nn}[x_i, x_o] = 0$ then the output $x_o$ does not depend on input $x_i$. Increasing the layers and hidden dimensions of the NNs will provide us with richer dependence function complexities.

**Representing categorical variables**. Assume that in the input $\mathbf{X}$, we have a column $X_c$ having $|C|$ different categorical entries. One way to handle categorical input is to do one-hot encoding on the column $X_c$ and end up with $|C|$ different columns, $X_c = [X_{c_1}, X_{c_2}, \cdots, X_{c_C}]$. We replace the single categorical column with the corresponding one-hot representation in the original data. The MLP capturing path dependencies $\mathbf{S}$ will need to be updated accordingly. Whatever connections where previously connected to the categorical column $X_c$ should be maintained for all the one-hot columns as well. Thus, we connect all the one-hot columns to represent the same path connections as the original categorical column.

## 3.3 Learning

Using the rich and compact functional representation achieved by using the 'neural' view, the learning task is to fit the neural networks to achieve the desired dependency structure $\mathbf{S}$, along with fitting the regression to the input data $\mathbf{X}$. Given the input data $\mathbf{X}$ we want to learn the functions as described by the NGMs 'graphical-view', Fig. 1. These can be obtained by solving the multiple regression

problems shown in neural view, Fig. 2. We achieve this by considering the neural view as a multi-task learning framework. The goal is to find the set of parameters $\{\mathcal{W}\}$ that minimize the loss expressed as the distance from $X_{\mathcal{I}}^k$ to $f_{\mathcal{W}}(X_{\mathcal{I}}^k)$ while maintaining the dependency structure provided in the input graph **G**. We can define the regression operation as follows:

$$\underset{\mathcal{W}}{\arg\min} \sum_{k=1}^{M} \left\| X_{\mathcal{I}}^k - f_{\mathcal{W}}(X_{\mathcal{I}}^k) \right\|^2 \tag{1}$$

$$s.t.\,(\Pi_i|W_i|) * S^c = 0$$

Here, $S^c$ represents the compliment of the matrix $S$, which essentially replaces 0 by 1 and vice-versa. The $A * B$ represents the hadamard operator which does an element-wise matrix multiplication between the same dimension matrices $A, B$. Including the constraint as a lagrangian term with $\ell_1$ penalty and a constant $\lambda$ that acts a tradeoff between fitting the regression and matching the graph dependency structure, we get the following optimization formulation

$$\underset{\mathcal{W}}{\arg\min} \sum_{k=1}^{M} \left\| X_{\mathcal{I}}^k - f_{\mathcal{W}}(X_{\mathcal{I}}^k) \right\|^2 + \lambda \left\| (\Pi_i|W_i|) * S^c \right\|_1 \tag{2}$$

Though the bias term is not explicitly written in the optimization to avoid cluttering, we learn the weights $\{W_i\}$ and the biases $\{b_i\}$ while optimizing for Eq. 2. In our implementation, the individual weights are normalized using $\ell_2$-norm before taking the product. We normalize the regression loss and the structure loss term separately, so that both the losses are on a similar scale while training and recommend the range of $\lambda$=[1e-2, 1e2]. Appropriate scaling is applied to the input data features.

**Proximal Initialization strategy**: To get a good initialization for the NN parameters $\mathcal{W}$ and $\lambda$ we implement the following procedure. We solve the regression problem described in Eqn. 1 without the structure constraint. This gives us a good initial guess of the NN weights $\mathcal{W}^0$. We choose the value $\lambda = \left\| (\Pi_i|W_i^0|) * S^c \right\|_2^2$ and update after each epoch. Experimentally, we found that this strategy may not work optimally in few cases and in such cases we recommend fixing the value of $\lambda$ at the beginning of the optimization. The value of $\lambda$ can be chosen such that it brings the regression loss and the structure loss values to same scale.

The learned NGM describes the underlying graphical model distributions, as presented in Alg. 1. There are multiple **benefits** of jointly optimizing in a multi-task learning framework modeled by the neural view of NGMs, eq. 2. First, sharing of parameters across tasks helps in significantly reducing the number of learning parameters. It also makes the regression task more robust towards noisy and anomalous data points. Second, we fully leverage the expressive power of the neural networks to model complex non-linear dependencies. Additionally, learning all the functional dependencies jointly allows us to leverage batch learning powered with GPU based scaling to get quicker runtimes.

### 3.4 INFERENCE

Inference is the process of using the graphical model to answer queries. Calculation of marginal distributions and conditional distributions are key operations for inference. Since NGMs are discriminative

---

**Algorithm 2:** NGMs: Inference algorithm

**Function** `gradient-based`$(f_{\mathcal{W}}, X_I)$**:**
  $\{X_k, X_u\} \leftarrow X_I$, split the data
  $X_k \leftarrow$ fixed tensor (known)
  $X_u \leftarrow$ learnable tensor (unknown)
  $f_{\mathcal{W}} \leftarrow$ freeze weights
  **do**
    $X_I \leftarrow \{X_k, X_u\}$
    $X_P = f_{\mathcal{W}}(X_I)$
    $\mathcal{L}_{\text{In}} = \|X_P[k] - X_I[k]\|_2^2$
    $X_u \leftarrow$ updated by backprop on $\mathcal{L}_{\text{In}}$
  **while** $\mathcal{L}_{In} > \epsilon$
  **return** $X_I$

**Function** `message-passing`$(f_{\mathcal{W}}, X^0)$**:**
  $X_K + X_U^0 \leftarrow X^0$, split the data
  $t = 0$
  **while** $\left\| X^t - X^{t-1} \right\|_2^2 > \epsilon$ **do**
    $\{X_u^t; X_k\} = f_{\mathcal{W}}(\{X_u^{t-1}; X_k\})$
    $t = t + 1$
  $X^t \leftarrow X_K + X_U^t$
  **return** $X^t$

**Function** `NGM-inference`$(f_{\mathcal{W}}, X^0)$**:**
  Input: $f_{\mathcal{W}}$ trained NGM model
  $X^0 \in \mathbb{R}^{D \times 1}$ (mean values for unknown)
  $X \leftarrow$ `message-passing`$(f_{\mathcal{W}}, X^0)$
    $\cdots$ or $\cdots$
  $X \leftarrow$ `gradient-based`$(f_{\mathcal{W}}, X^0)$
  **return** $X$

---

models, for the prior distributions, we follow the frequentist approach and directly calculate them from the input data. We consider two iterative procedures to answer conditional distribution queries over NGMs described in Alg. 2. We split the input data $X_k + X_U \leftarrow X$ into two parts, $k$ denotes the known (observed) variable values and $u$ denotes the unknown (target) variables. The inference task is to predict the values of the unknown nodes based on the trained NGM model distributions. In the fist approach, we use the popular message passing algorithms that keeps the observed values of the features fixed and iteratively updates the values of the unknowns until convergence. We developed an alternative algorithm which is efficient and is our recommended approach to do inference in NGMs.

**Gradient based approach**: The weights of the trained NGM model are frozen once trained. The input data is divided into fixed $X_k$ (observed) and learnable $X_u$ (target) tensors. We then define a regression loss over the known attribute values as we want to make sure that the prediction matches values for the observed features. Using this loss we update the learnable input tensors until convergence to obtain the values of the target features. Since the NGM model is trained to match the output to the input, we can view this procedure of iteratively updating the unknown features such that the input and output matches. Based on the convergence loss value reached after the optimization, one can assess the confidence in the inference. Furthermore, plotting the individual feature dependency functions also helps in gaining insights about predicted values.

**Obaining probability distributions.** It is often desirable to get the full probability density function rather than just a point value for any inference query. In case of categorical variables, this is readily obtained as we output a distribution over all the categories. For real or numerical features, we consider a binned input on the input side and real value output. In this case, the regression term of the loss function, Eq. 3 will take binned input and output a real value for the real valued features $\sum_{k=1}^{M} \left\| X_{\mathcal{I}\text{-real}}^k - f_{\mathcal{W}}(\text{proj}(X_{\mathcal{I}\text{-binned}}^k)) \right\|^2$. In practice, given a distribution over different categories obtained during the NGM inference, we clip the individual values between $[\epsilon, 1]$ and then divide by the total sum to get the final distribution.

---

**Algorithm 3:** NGMs: Sampling algorithm

**Function** get-sample($f_{\mathcal{W}}, \mathcal{D}_s$):
    $D = \text{len}(\mathcal{D}_s)$
    $X \in \mathbb{R}^{D \times 1}$ (random init, learnable tensor)
    Sample $1^{st}$ feature value from empirical
      marginal distribution $x_1 \sim \mathcal{U}(P(x_1))$
    **For** $i = 2, \cdots, D$ **do**
        $X_k \leftarrow X[1 : i-1]$ (fixed tensor)
        $X_u \leftarrow X[i : D]$ (learnable tensor)
        $X \leftarrow \{X_k, X_u\}$
        $X \leftarrow \text{NGM-inference}(f_{\mathcal{W}}, X)$
        $X[i] \sim \mathcal{U}(P(x_i | X[1 : i-1]))$
    **return** $X$

**Function** NGM-sampling($f_{\mathcal{W}}, \boldsymbol{G}$):
    Input: $f_{\mathcal{W}}$ trained NGM model
    Randomly choose $x_i$'th feature
    $\mathcal{D}_s$=BFS($\mathbf{G}, x_i$) [undirected]
    $\cdots$ queue the features $\cdots$
    $\mathcal{D}_s$=topological-sort($\mathbf{G}$) [DAGs]
    $X \leftarrow$ get-sample($f_{\mathcal{W}}, \mathcal{D}_s$)
    **return** $X$

---

### 3.5 SAMPLING

One common way of sampling is to define cumulative density functions and then sample from them. This will not be possible for NGMs. So, instead, we propose a procedure akin to Gibbs sampling as described in Alg. 3.

We based our sampling procedure to follow $X_i \sim \mathcal{U}(f_{nn}(\text{nbrs}(X_i)))$. Note that $\text{nbrs}(X_i)$ will be $MB(X_i)$ for DAGs. We start sampling by choosing a feature at random. To get the order in which the features will be sampled, we do a Breadth-first-search (topological sort in DAGs) and arrange the nodes in $\mathcal{D}_s$. In this way, the immediate neighbors are chosen first and then the sampling spreads over the graph away from the starting feature. As we go through the ordered features in the sampling procedure, we sample the value of each feature from the conditional distribution based on previously assigned values and then keep it fixed for the subsequent iterations (feature is now observed). We then call the inference algorithm conditioned on these fixed features to get the distributions over the unknown features. This process is repeated till we get a sample value of all the features.

Our sampling procedure differs from the Gibbs sampling with regards to conditional distribution calculations. Traditionally, in Gibbs sampling, sample $X^k$ is derived from the previous sample $X^{k-1}$ by following a conditional distribution update. Specifically, the value of $X_i^k$ is obtained according to the distribution specified by $p\left(X_i^k | X_1^k, \cdots, X_{i-1}^k, X_{i+1}^{k-1}, \cdots, X_D^{k-1}\right)$. The new sample of the NGM is not derived from the previous sample, hence we avoid the 'burn-in' period issue with Gibbs

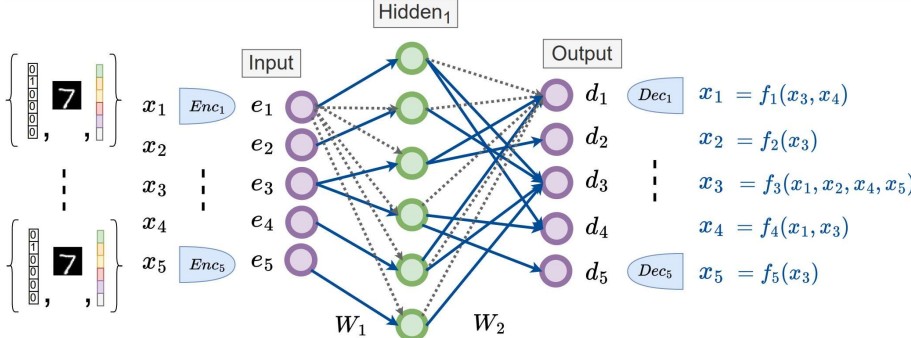

Figure 3: **Neural view with Projection modules of NGMs**: The input **X** can be one-hot (categorical), image or in general an embedding (text, audio, speech and other data types). Projection modules (encoder + decoder) are used as a wrapper around the neural view of NGMs. The architecture choice of the projection modules depends on the input data type and users' design choices. Note that the output of the encoder can be more than 1 unit ($e_1$ can be a hypernode). In that case, we just need to adjust the graph dependency structure **S** to account for that many units and the corresponding feature connections. Same will be the case with the decoder side of the architecture. The remaining details are similar to the ones described in Fig. 2

sampling where one has to ignore the initial set of samples. The conditional updates for the NGMs are of the form, $p\left(X_i^k, X_{i+1}^k, \cdots, X_D^k | X_1^k, \cdots, X_{i-1}^k\right)$. We keep on fixing the value of features and run inference on the remaining features until we have obtained the values of all the features and thus get a new sample. The inference algorithm of the NGM facilitates conditional inference on multiple unknown features over multiple observed features. We leverage this capability of the inference algorithm for faster sampling from NGMs.

### 3.6 Extension to generic data types

The learning, inference and sampling algorithms proposed for NGMs in the previous section can be extended to any generic input data type. This implies that the data **X** can be real, categorical, image or have an embedding based representation. We add a Projection module consisting of an encoder and decoder that act as a wrapper around the neural view of the NGMs. With a slight modification, we obtain the following optimization for generic data types,

$$\underset{\mathcal{W},\text{proj}}{\arg\min} \sum_{k=1}^{M} \left\| X_{\mathcal{I}}^k - f_{\mathcal{W}}(\text{proj}(X_{\mathcal{I}}^k)) \right\|^2 + \lambda \left\| (\Pi_i|W_i|) * S^c \right\|_1 \tag{3}$$

The Projection module can be jointly learned in the optimization, as shown in Eq. 3, or one can add fine-tuning layers to the pretrained versions depending on the data type and user preference.

Alternatively, one can extend the idea of soft-thresholding the connection patterns to the encoder and decoder networks. This will result in an efficient training strategy that leverages batch processing.

$$\underset{\mathcal{W}^n, \mathcal{W}^e, \mathcal{W}^d}{\arg\min} \sum_{k=1}^{M} \left\| X_{\mathcal{I}}^k - f_{\mathcal{W}}(X_{\mathcal{I}}^k) \right\|^2 + \lambda_n \left\| (\Pi_i|W_i^n|) * S_n^c \right\|_1 \tag{4}$$
$$+ \lambda_e \left\| (\Pi_i|W_i^e|) * S_e^c \right\|_1 + \lambda_d \left\| (\Pi_i|W_i^d|) * S_d^c \right\|_1$$

where, the connectivity of the input $x$ and the input to the neural view is modeled by the $\ell_1$ sparsity term for the encoder network's sparsity pattern $S_e^c$. Similar procedure is applied to the decoder side.

If the Projection modules are used, the number of nodes in the neural view input should be adjusted according to the output units of the encoder. Similar adjustment is needed for neural view output and the decoder. In real world applications, we often find inputs consisting of mixed datatypes. For instance, in the gene expression data, there can be additional meta information (categorical) or images associated with the genes. Optionally, one can desire to utilize node embeddings from some other pretrained deep learning models. NGMs are designed to handle such **mixed** input data types simultaneously which are otherwise very tricky to accommodate in the existing graphical models.

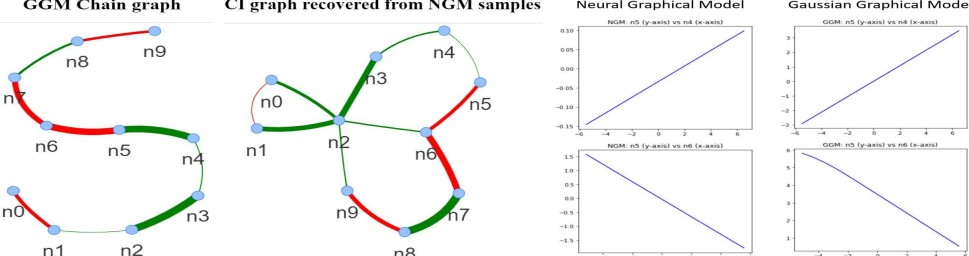

Figure 4: The leftmost graph shows the chain graph **G** (partial correlations in green are positive, red are negative, thickness shows the correlations strength) obtained from the initialized partial correlation matrix. Samples $\mathbf{X} \in \mathbb{R}^{2000 \times 10}$ were drawn from the GGM. NGM was learned on the input $(\mathbf{X}, \mathbf{G})$. The 2 plots on the right show the dependency functions of NGM and GGM for a particular node by varying its neighbor's values. The positive and negative correlations are reflected in the slope of the curve, as expected analytically. We then sampled from the learned NGM to obtain data $\mathbf{Xs} \in \mathbb{R}^{M_s \times 10}$. The graph, second from the left, shows the recovered graph by running `uGLAD` Shrivastava et al. (2022a) on $\mathbf{Xs}$. We can observe that it missed some of the edges but most of the connections along with the correlations signs were retrieved from the NGM samples.

## 4 EXPERIMENTS

We evaluate NGMs on synthetic and real data. Appendix A contains some best practices that we developed while working with NGMs. In Appendix B, we present an analysis of CDC's **Infant Mortality** Data (of Health et al.) using NGMs, which highlights NGMs-generic architecture's ability to model mixed input datatypes.

### 4.1 MODELING GAUSSIAN GRAPHICAL MODELS

We designed a synthetic experiment to study the capability of NGMs to represent Gaussian graphical models. The aim of this experiment is to see (via plots and sampling) how close are the distributions learned by the NGMs to the GGMs.

Table 1: The recovered CI graph from NGM samples is compared with the CI graph defined by the GGMs precision matrix. Area under the ROC curve (AUC) and Area under the precision-recall curve (AUPR) values for 10 runs are reported, refer to Fig. 4.

| Samples | AUPR | AUC |
|---------|------|-----|
| 1000 | $0.84 \pm 0.03$ | $0.91 \pm 0.002$ |
| 2000 | $0.86 \pm 0.02$ | $0.93 \pm 0.001$ |
| 4000 | $0.96 \pm 0.00$ | $0.99 \pm 0.003$ |

**Setup**: *Define the underlying graph.* We defined a 'chain' (or path-graph) containing D nodes as the underlying graph. We chose this graph as it allows for an easier study of dependency functions.

*Fit GGM and get samples.* Based on the underlying graph structure, we defined a precision matrix $\Theta$ that randomly samples its entries from $\Theta_{i,j} \sim \mathcal{U}\{(-1, -0.5) \cup (0.5, 1)\}$. We then used this precision matrix as a multivariate Gaussian distribution parameter to obtain the input sample data $\mathbf{X}$. We get the corresponding partial correlation graph $\mathbf{G}$ by using the formula, $\mathrm{P}_{X_i, X_j} . \mathbf{X}_{D \setminus i,j} = -\frac{\Theta_{i,j}}{\sqrt{\Theta_{i,i} \Theta_{j,j}}}$.

*Fit NGM and get samples.* We fit a NGM on the input $(\mathbf{X}, \mathbf{G})$. We chose $H = 30$ with 2 layers and non-linearity `tanh` for the neural view's MLP. Training was done by optimizing eq. 2 for the input, refer to Fig. 4. Then, we obtained data samples $\mathbf{Xs}$ from the learned NGM.

**Analysis**: *'How close are the GGM and NGM samples?'* We recover the graph using the graph recovery algorithm `uGLAD` on the sampled data points from NGMs and compare it with the true CI graph. Table 1 shows the graph recovery results of varying the number of samples from NGMs. We observe that increasing the number of samples improves the graph recovery, which is expected.

*'Were the NGMs able to model the underlying distributions?'* The functions plot (on the right) in Fig. 4 plots the resultant regression function for a particular node as learned by NGM. This straight line with the slope corresponding to the partial correlation value is what we expect theoretically for the GGM chain graph. This is also an indication that the learned NGMs were trained properly and reflect the desired underlying relations. Thus, NGMs are able to represent GGM models.

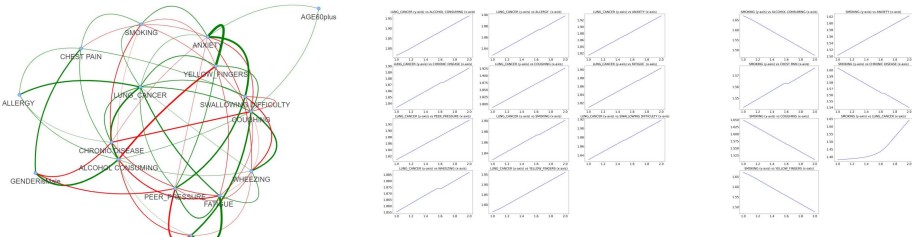

Figure 5: (left) The CI graph recovered by `uGLAD` for the Lung cancer data. Plots on the right show the conditional distribution for the features P(Lung cancer='Yes'| nbrs(Lung cancer)) and P(Smoking| nbrs(Smoking)) based on their neighbors. We used a 2-layer NGM with hidden size $H = 30$ and non-linearity as `tanh`. NGMs are able to capture the non-linear dependencies between the features. Interestingly the NGMs match the relationship trends discovered (positive and negative correlations) by the corresponding CI graph.

## 4.2 LUNG CANCER DATA ANALYSIS

We analysed a lung cancer data on Kaggle using NGMs. The effectiveness of cancer prediction system helps people to know their cancer risk with low cost and it also helps people to take appropriate decisions based on their cancer risk status. This data contains 284 instances of patients and for each patient 16 features (Gender, Smoking, Anxiety, Lung cancer present, etc.) are collected. Each entry is a binary entry (YES/NO) or in some cases (AGE), entries are binarized. Particularly, we used NGMs to study how different features are related and discover their underlying functional dependencies.

The input data along with the CI graph recovered using `uGLAD` were used to learn a NGM in Fig. 5. In order to gauge the regression quality of NGMs, we compare with logistic regression to predict the probability of feature values given the

| Methods | Lung-cancer | Smoking |
|---------|-------------|---------|
| LR | $0.95 \pm 0.02$ | $0.71 \pm 0.01$ |
| NGM | $0.96 \pm 0.01$ | $0.79 \pm 0.02$ |

Table 2: 5-fold CV results.

values of the remaining features. Table. 2 shows regression results of logistic regression (LR) and NGMs on 2 different features, 'lung cancer' & 'smoking'. The prediction probability for NGMs were calculated by running inference on each test datapoint, eg. P(lung-cancer='yes'| $f_i = v_i \, \forall i$ in test data). This experiment primarily demonstrates that a single NGM model can robustly handle fitting multiple regressions and one can avoid training a separate regression model for each feature while maintaining at-par performance. Furthermore, we can obtain the dependency functions that bring in more interpretability for the predicted results, Fig. 5. Samples generated from this NGM model can be used for multiple downstream analyses.

## 5 CONCLUSIONS

This work attempts to improve the usefulness of probabilistic graphical models by extending the range of input data types and distribution forms such models can handle. Neural Graphical Models provide a compact representation for a wide range of complex distributions and support efficient learning, inference and sampling. The experiments are carefully designed to systematically explore the various capabilities of NGMs. Though NGMs can leverage GPUs and distributed computing hardware, we do forsee some challenges in terms of scaling in number of features and performance on very high-dimensional data. Using NGMs for images & text based applications will be interesting to explore. We believe that NGMs is an interesting amalgam of the deep learning architectures' expressivity and Probabilistic Graphical models' representation capabilities.

**Upcoming version: Discovering the dependency graph with NGMs.** We are currently working on a version of NGM that can jointly discover the feature dependency graph along with fitting the regression. One way can be to optimize this loss function,

$$\underset{\mathcal{W}, \text{proj}}{\arg\min} \sum_{k=1}^{M} \left\| X_{\mathcal{I}}^k - f_{\mathcal{W}}(\text{proj}(X_{\mathcal{I}}^k)) \right\|^2 + \lambda \left\| (\Pi_i |W_i|) * S_{\text{diag}} \right\|_1 \tag{5}$$

where $S_{\text{diag}}$ has diagonal entries as 1. Essentially, we start with a fully connected graph and then the $\ell_1$ term induces sparsity. This will be helpful in cases where input **G** is not provided.

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
