# OpenReview forum: "Neural Graphical Models"
_ICLR.cc/2023/Conference — Submitted to ICLR 2023_

### Official Review · Reviewer_6f9Q · 2022-10-16

**Confidence:** 3
**Correctness:** 2
**Technical Novelty And Significance:** 3
**Empirical Novelty And Significance:** 2
**Recommendation:** 3

**Clarity, Quality, Novelty And Reproducibility:**

The paper is fairly clearly written, subject to confusion that may just be the result of my unfamiliarity with neural networks.  If anything, I think they spend too much space discussing issues that most experts would consider to be trivial (e.g. one-hot encoding, Lagrangian for optimization).

So far as I can tell, the overall contribution is novel, though (partly due to a lack of expertise) it is hard for me to state exactly what it is.

I imagine the results are reproducible, though I couldn't get all of the supplied Jupyter notebook to run.


**Strength And Weaknesses:**

The paper is fairly clearly written, and comes with code that appears to run OK.  (Though there is a variable `model_NGM_trained` that is not supplied and breaks the attached Jupyter Notebook.)

The main weakness in my view is the lack of experiments comparing the method in the paper to the many others that the authors mention.

The only comparison that is made is learning a graph and its distribution from multivariate Gaussian data.  The recovered graph (done with uGLAD, an existing method) introduces three false edges, and the dependencies that the method in this paper discovers have the correct sign, but the magnitude of the dependence (seen in the thickness of edges) is very poorly recovered.  The accompanying plots are so small that the **massively** different scales on the y-axes are hard to see (and they are only given for two pairs of variables chosen by the authors).  This is generally not a good sign, and given that the sample size is 2000 it seems a little worrying (as does the edge between nodes 0 and 2, which is comparatively strong).

The claims made about speed and flexibility are not really supported in any experiments; no runtimes are given at all.  There are also not any experiments for inference or sampling with directed graphs, in spite of claims being made that this method can also deal with them.

A lot of statements are made about benefits of joint minimization that are not supported by evidence (neither simulations nor citations).

My **main feedback** is: do more comparisons (you can put some in the appendix) and give evidence for your claims.

**Summary Of The Paper:**

This paper introduces a method of learning, performing inference in, and simulating from graphical models (undirected, directed, or a combination) using neural networks.  The authors claim that this greatly increases the flexibility while keeping computational complexity relatively low.

I have some expertise in graphical models, but limited experience with neural networks, some of the comments below may be erroneous.

**Summary Of The Review:**

Overall, there just isn't enough here for me to recommend that the paper be accepted.  The method might be very good, but the authors haven't presented any evidence that it is, and the little they do provide suggests that it might not be that great.


**Other Coments**

Page3: "conditional independence graphs...are a type of graphical models that are based on the multivariate Gaussian distribution." - This is not true, conditional independence graphs can represent any distribution, including non-parametric models.


_Minor Points_
Minor Points:
 - There are quite a lot of places where references should be in parentheses but are not.
 - page 3 - after (Sutton et al., 2012) missing a space.
 - "$S^c$ essentially replaces 0 by 1 and vice versa" - be precise here please.  If it does this other than on the diagonal, then say so.
 - In the next paragraph, "Hadamard" and "Lagrangian"
 - Equation (2) seems a bit basic.
 - don't write [1e-2, 1e2] but rather $[10^{-2}, 10^2]$.
 - page 6: "fist" -> "first"
 - "we clip the individual values between $[\epsilon, 1]$": what is $\epsilon$?
 - Why is it not possible to use CDFs with neural networks? (e.g. Chilinski and Silva, 2020)
 - page 6: "till" -> "until"
 - page 8: "we randomly sample entries in the precision matrix between $(-1, -0.5) \cup (0.5, 1)$..." How do you guarantee positive definiteness?  Does this work precisely because the graph is a chain?
 - page 9: "a lung cancer data on Kaggle" -> "a lung cancer dataset on Kaggle"
 - page 9: please make your plots bigger; put most of them in the appendix and select a few to have in Figure 4.
 - several places on page 9: can you please write probabilities in mathmode?
 - Please ensure that the capitalization in your references is correct.  You can get it right by adding {} around words that are correctly written in the bibtex-entry.

---

### Official Review · Reviewer_1gso · 2022-10-24

**Confidence:** 4
**Correctness:** 2
**Technical Novelty And Significance:** 2
**Empirical Novelty And Significance:** 2
**Recommendation:** 3

**Clarity, Quality, Novelty And Reproducibility:**


**Some important claims are not supported**

    "Generative models have higher bias - they make more assumptions about the form of the distribution."

I don't get why this should be the case. Given the same graphical
structure over X and Y, the parameters can be estimated for either
P(X,Y) or P(Y|X), how is the latter making less assumptions?

---

    "These PGMs [including PCs] often make simplifying assumptions on the underlying
    distribution and have certain restrictions on the input data type
    that can be handled".

Probabilistic Circuits can handle numerical (both continuous and
discrete) and categorical data (see e.g. [1]).  In terms of distributional
assumptions, they are also quite flexible.  Which assumption made by
PCs is relaxed in NGMs?

---

**The presentation of the proposed approach should be improved**

I think that the proposed approach is not presented in a principled
way. Since NGMs are probabilistic models, at mimimum I
would expect to read:

- What is the parametric form of the joint distribution encoded by a NGM?

- How is it guaranteed that the model in Fig.2 is a probability distribution?

- How is learning based on probability theory?

---

**Relevant work in this area is not considered**

The paper should at least cite related work like:

- probabilistic models for hybrid categorical/numerical distributions (e.g. Mixed SPNs [1], Manichean models [2])
- unifying PGMs and neural networks (e.g. see [3], [4])

---

**The experimental section is weak**

The experimental section doesn't provide evidence of the merits of
the proposed approach. I would suggest to explicitly write the
research questions supported by the empirical evaluation.
The main motivation behind NGMs is their flexibility and lack of distributional assumptions, yet the experiments feature a scenario with Gaussians only and a scenario with categorical (binary) variables. Why?
Where appropriate, the experiments should include a comparison with related approaches (see above).

---

References:

[1] Molina, Alejandro, et al. "Mixed sum-product networks: A deep architecture for hybrid domains." AAAI, 2018.

[2] Yang, Eunho, et al. "Mixed graphical models via exponential families." Artificial intelligence and statistics. PMLR, 2014.

[3] Li, Chuan, and Michael Wand. "Combining markov random fields and convolutional neural networks for image synthesis." CVPR. 2016.

[4] Johnson, Matthew J., et al. "Composing graphical models with neural networks for structured representations and fast inference." NeurIPS 2016.


**Strength And Weaknesses:**

Strengths:
+ Hybrid continuous/discrete probabilistic modelling has application in many scientific areas
+ I think that parametrizing the factors of a PGM with neural networks is a sensible idea

Weaknesses (see section below for details):
- Some important claims are not supported
- The presentation of the proposed approach should be improved
- Relevant work in this area is not considered
- The experimental section is weak

**Summary Of The Paper:**

This paper proposes a novel class of probabilistic models, dubbed Neural Graphical Models (NGMs), which parametrize the distribution with neural networks. This setup supports mixed continuous/discrete distributions and arbitrary graphical structures.

**Summary Of The Review:**

While this work is addressing a relevant problem and the key idea behind NGMs is reasonable, I cannot recommend the current version of the paper for publication.

---

### Official Review · Reviewer_3Bs7 · 2022-10-31

**Confidence:** 4
**Correctness:** 2
**Technical Novelty And Significance:** 2
**Empirical Novelty And Significance:** 2
**Recommendation:** 3

**Clarity, Quality, Novelty And Reproducibility:**

The portion of the paper related to conditional inference is completely unclear. I was unable to understand the idea how this works as mentioned above.

Other than the issue of conditional sampling the rest of the paper is very well written and quite easy to follow. The experiments as described don't rely to conditional sampling, so they appear to be reproducible as well.

The authors do claim a lot more than they have provided evidence for in this paper which detracts from the quality of the paper. For example they claim to support categorical variables and images as random variables. Although they do explain theoretically how to support these variables there is no empirical confirmation.


**Strength And Weaknesses:**

The paper has clearly taken on an ambitious problem which attempts to unify generative and discriminative modeling with the efficiency of neural network inference. They provide a very intuitive representation of the graphical model in a neural network and devise easy to comprehend objective functions that are used for training the neural networks.

The paper doesn't seem to have hit the mark in terms of representing the uncertainty in a probabilistic graphical model. The description in the paragraph titled "Obtaining probability distributions" is completely unclear. How do you get a probabilistic output by inputting a binned value and what is a binned value in any case. The section on sampling, 3.5, describes drawing a full joint distribution and not really a conditional distribution. Even this description is somewhat unclear. The line, "we sample the value of each feature from the conditional distribution based on previously assigned values" is not clear. I understand how we can draw the value of one feature from the previously assigned value, but how do we get the conditional distribution of a feature? None of the experiments seem to cover the case of conditional distribution. Showing that joint samples retain pairwise correlations, Figure 5, is reassuring but doesn't prove anything about conditional distributions and neither does the accuracy results on a regression task, Table 2.


**Summary Of The Paper:**

The paper provides an approach to represent a probabilistic graphical model of relationships between features and a dataset over the same features using a neural network. This neural network can then be used draw samples from the joint distribution of the features, and it can be used to do conditional inference.

The contributions of the paper include the neural network learning algorithm, an algorithm to infer the values of unknown variables given some observed variables, and an algorithm to draw samples from the graph. The work is evaluated on a synthetic dataset by comparing pairwise correlations of sampled variables and on kaggle dataset by comparing to logistic regression.



**Summary Of The Review:**

Nice tidy idea for representing PGMs in a neural network, but a lot of unclear or unsupported claims around representing uncertainty and mixed datatypes.

---

### Official Review · Reviewer_ZfGg · 2022-10-31

**Confidence:** 4
**Correctness:** 2
**Technical Novelty And Significance:** 3
**Empirical Novelty And Significance:** 2
**Recommendation:** 3

**Clarity, Quality, Novelty And Reproducibility:**

The idea seems to be somewhat novel. There has been previous work on using deep neural networks to represent complex local probability distributions. This paper seems to learn a global model that encodes structural dependencies using a single neural network. The experiments are weak at this point (see comments below) but seem to be reproducible. The paper is somewhat clear except most of the graphs in the experiments section were illegible. The quality of writing could be improved.

**Strength And Weaknesses:**

Strengths:
A new representation to encode probability distributions using neural networks.
The model has the capacity to encode various types of dependencies and data types.
The authors have described both learning and inference algorithms.
Weaknesses:
Weak experimental evaluation.


**Summary Of The Paper:**


In this paper the authors propose a new probabilistic graphical model representation called neural graphical models (NGMs). NGMs encode the joint probability distributions over random variables using multi-layer perceptrons. An NGM takes as input data and a graph that represents the probabilistic dependencies among the random variables. The graph structure is often provided by a domain expert or learned from the data itself.  A regression loss is minimized subject to the constraint that if xi and xj are neighbors in the input graph, then there is a path of non-zero weights from input xi to the corresponding output xj in the NGM.  As a result of such weight matrices, the outputs of NGMs represent regression functions over those input variables that are direct neighbors of the output variable. The main motivation around this representation is to be able to use a single representation that can accommodate various types of random variables as well as structural dependencies such as directed, undirected or mixed types. The authors describe inference and sampling procedures for these models and empirically test their performance on recovering a gaussian graphical model and on predicting lung cancer from a real world dataset.


**Summary Of The Review:**

I found the empirical evaluations for this paper to be insufficient. A large body of work exists for learning various types of probabilistic graphical models and proposing a general representation like NGMs that can support all kinds of dependencies and data types as well as being efficiently learnable and queryable, requires more analysis. The authors have presented results on recovering a simple gaussian graphical model. These results don't seem to empirically validate the claim that these models can be very expressive and represent rich and complex distributions. An important question also arises on how the number of hidden layers affect the accuracy of the recovered distributions.

The lung cancer results seems to compare logistic regression accuracy to MLP accuracy on predicting cancer. The results are quite obvious. Unless more sophisticated inference tasks are carried out using NGMs, its hard to understand their expressive power.
The scenarios in which an NGM would be a better choice than a Bayesian network or a Markov network is not quite well understood. For example, Given the structure of a discrete Bayesian network, one can efficiently learn its parameters if the model has a bounded number of parents. Why would one use an NGM in such scenarios?

I am not sure why the authors have mentioned Gibbs sampling. The algorithm they mention generates samples in order similar to importance sampling (see Gogate, "Sampling Algorithms for Probabilistic Graphical Models with Determinism", PhD Thesis, 2009).

Some other very important relevant work are:
Uria, Benigno, et al. "Neural autoregressive distribution estimation." The Journal of Machine Learning Research 17.1 (2016): 7184-7220.
Uria, Benigno, Iain Murray, and Hugo Larochelle. "RNADE: The real-valued neural autoregressive density-estimator." Advances in Neural Information Processing Systems 26 (2013).

---

### Decision · Program_Chairs · 2023-01-20

**Decision:**

Reject

**Justification For Why Not Higher Score:**

N/A

**Justification For Why Not Lower Score:**

N/A

**Metareview: Summary, Strengths And Weaknesses:**

Unanimous reviewer decision and no author response.